# Transcriptomic Analysis of Liver Tissue of Black Sea Bass (*Centropristis striata*) Exposed to High Nitrogen Environment

**DOI:** 10.3390/genes14071440

**Published:** 2023-07-13

**Authors:** Bingjian Liu, Xun Jin, Kun Zhang, Yifan Liu, Shaowen Wang, Shiyi Chen, Shufei Zhang, Xiaolong Yin

**Affiliations:** 1Marine Science and Technology College, Zhejiang Ocean University, Zhoushan 316021, China; liubingjian@zjou.edu.cn (B.L.); j782289657@163.com (X.J.); 15034676496@163.com (K.Z.); csy9915@126.com (S.C.); 2College of Marine Sciences, South China Agricultural University, Guangzhou 510642, China; wangsw@scau.edu.cn; 3Guangdong Provincial Key Laboratory of Fishery Ecology and Environment, South China Sea Fisheries Research Institute, Chinese Academy of Fisheries Sciences, Guangzhou 510300, China; zhangshufei@scsfri.ac.cn; 4Zhoushan Fisheries Research Institute of Zhejiang Province, Zhoushan 316021, China; xlyhndx@163.com

**Keywords:** *Centropristis striata*, path enrichment analysis, differentially expressed genes, nitrate stress, aquaculture

## Abstract

The black sea bass, *Centropristis striata*, is a potential candidate for commercial aquaculture. Due to inadequate removal of nitrogen in its breeding environment, *C. striata* exhibits increased nitrate concentration, which can cause acute toxicity, including energy metabolism damage and tissue damage. Therefore, RNA-seq technology was applied to characterize genes associated with toxicity tolerance under nitrate stress. The nitrate treatment caused significant changes in a total of 8920 genes, of which 2949 genes were up-regulated and 5971 genes were down-regulated. It was found that significantly enriched GO terms and KEGG were associated with blood microparticles, inhibitors of enzyme activity, and complement and coagulation cascade pathways. Furthermore, through bioinformatics analysis, it was found that these different pathways obtained in GO and KEGG enrichment analysis were mostly related to the immune and inflammatory response of fish. This study expands our understanding of the mechanism of nitrate stress affecting the liver function of *C. striata*.

## 1. Introduction

The black sea bass (*Centropristis striata*) is a species of Perciformes in the family Serranidae. It is a valuable marine fish with high consumption and strong adaptability. *C. striata* is also an important species in American commercial and recreational fisheries [1]. *C. striata* inhabits continental shelf waters of the eastern US from Florida to Maine and feeds on fish and decapods. *C. striata* has a strong, fusiform body with neatly arranged ctenoid scales, resembling a string of pearls, so it is known for its pearl spots. It is also known as the emerald spots fish because of the eight spots on the adult’s head and fluorescent stripes on both sides of the body. Its beautiful body and color make it a famous ornamental fish. From 1980 to 1990, the total catch in the South Atlantic averaged about 2.5 million pounds, and the highest catch in 1988 was 3.6 million lbs. The average catch from 2000 to 2010 was 1.15 million lbs, and the highest was 1.75 million lbs [2]. This indicates a strong demand for the species, but it is strictly regulated by various management measures, such as seasonal fishing bans, and, since 2006, the total allowable catch or annual catch limit [3]. The University of North Carolina at Wilmington (UNCW) conducted research on *C. striata* aquaculture, which showed that *C. striata* can be reproduced reliably in captivity [4,5,6], achieve systematic cultivation from egg to adult stage and win a lucrative sales market. *C. striata* was introduced into China in 2002 for artificial breeding because of its strong disease resistance, fast growth, fewer spines, and great market potential, which make it an excellent variety suitable for aquaculture. Some excellent biological breeding characteristics show that it is easy to breed, has a high yield, and can obtain high economic benefits at a low breeding cost. At present, *C. striata* breeding and growing technologies have initially taken shape.

However, with the continuous expansion of the scale of fishery aquaculture, there are inevitably many problems, such as the deterioration of the water quality by sewage runoff [7], the problem of increasing the rate of fish mass growth, and death from fish body infections, which directly affect the production efficiency of aquaculture. Therefore, more and more attention has been paid to improving the stress resistance of cultured fish [8]. In the aquaculture process, more inexpensive filtration equipment has been used to treat aquaculture wastewater to reduce costs. The main pollutants in the water include dissolved or suspended particulate matter and fish excrement that is not cleaned in time [9]. The utilization rate of fish feeding was low, and only a small part of the feeding could effectively be used by farmed fish; most of their feed is expelled as excrement [10]. Manure that is not cleaned up in time contains high levels of organic matter that consumes a lot of oxygen when it decomposes in water. In this environment, harmful substances such as ammonia nitrogen and nitrite rapidly accumulate, deteriorating the water quality [8]. Ammonia, nitrite and nitrate nitrogen compounds in aquaculture environments easily reach toxic levels, affecting the normal growth and health of aquatic animals as a limiting factor [11].

Currently, the increase in nitrite concentration has gradually become a key issue in high-density aquaculture systems [12]. Increased levels of nitrite can cause acute toxicity in aquatic animals, including alteration of energy metabolism and tissue damage [13]. Nitrite reduces the oxygen-carrying capacity of the blood by oxidizing the iron in hemoglobin to a trivalent state [14]. Nitrate is the main nitrogen-containing component produced by nitrification, but compared with ammonia and nitrite, its toxicity to aquatic animals is generally not considered to be fatal [15]. High concentrations of nitrate can cause aquatic animal poisoning and even death due to the final production of ammonia oxidation to nitrate [16,17,18]. Because of the relatively low level of nitrate in the aquaculture environment, the current research on nitrate toxicity of marine fish is still insufficient [11]. Unfortunately, in circulating aquaculture systems and biological flocculation technology systems, nitrates can reach toxic or lethal levels [19,20].

In recent years, with the development and expansion of sequencing technology, gene function research, and functional annotation libraries, transcriptome analysis based on RNA sequencing has become more convenient and economical, providing a good way to explore the stress-regulated expression of non-model species [21]. Based on the principle of similarity, functional annotation refers to multiple annotation libraries that have greatly promoted omics research, effectively saved time cost, and effectively applied open-source gene annotation information from all over the world. Moreover, they have significantly improved the accuracy of functional annotation through the comparison of different libraries and provided data support for downstream problem solving [22,23,24]. Functional annotation has been widely used in aquaculture to identify candidate regulatory genes involved in growth, development, reproduction, disease, immunity, stress, and toxicology [25]. It is helpful to understand the possible defense regulatory pathways and adaptive network pathways of fish in adverse environments and beneficial to explore the key regulatory genes and improve the related problems encountered in aquaculture. At present, this technology has been well used in some cultured fish such as Ctenopharyngodon idella [26], Betta splendens [27] and Cyprinus carpio [28].

The transcriptome of an organism can be obtained through next-generation RNA sequencing technology [29], which is preferred to cataloging and quantifying the entire set of transcripts for specific tissues, developmental stages, or physiological conditions in response to specific stresses [30]. Due to the use of recirculating aquaculture systems and the inability to clean up fish bait residues and manure, *C. striata* aquaculture has long been suffering from nitrate stress. Therefore, the purpose of this research is to find differential expression data through RNA-seq technology and explore a series of problems caused by the increase of nitrate concentration in *C. striata* breeding. At the same time, our data provide valuable genomic resources for *C. striata*.

## 2. Materials and Methods

### 2.1. Ethics Committee Permit and Experimental Procedure Compliance

All procedures in this study were approved by the Animal Ethics Committee of Zhejiang Ocean University (Zhoushan, China), and carried out in accordance with the Regulations for the Administration of Laboratory Animals (Decree No. 2 of the State Science and Technology Commission of the People’s Republic of China, 14 November 1988). All experimental operations and procedures are legal and compliant. All experiments were conducted in accordance with the reference experimental guidelines and regulations, and the US National Research Council’s guidelines for the Use and Care of Laboratory Animals were followed. From animal transportation to the experiment itself and its completion, there were no incidents of abuse or other infractions; thus, the experiment met ethical requirements.

### 2.2. Sample Extraction

The experimental samples came from adult black sea bass from a fish farm in Zhoushan, Zhejiang Province. We collected six adult individuals for experiments. Among them, three were randomly selected as the experimental group, and the remaining as the control group. The *C. striata* were transported to the laboratory using plastic containers and sea water. No animals died during the transportation. In the laboratory, the sample was divided into the experimental group (nitrate concentration 450 mg/L) and control group (naturally filtered seawater was maintained below 1 mg/L). The experimental fish were kept temporarily for one week, during which the two groups were cultured under the same conditions with the breeding temperature set at 24 °C; the salinity was controlled at about 26, and the aeration was maintained in the tank, while at the same time, the water was changed once a day. The experiment was conducted a week later, and a concentration of nitrate (sodium nitrate) was added to the water of the experimental group and adjusted to 450 mg/L. During the experiment, the *C. striata* were fed normally with no further changes, except that the new water (experimental group) was adjusted to the experimental nitrate concentration before changing. The control and the experimental group were cultured for 14 days, and euthanasia was performed with 300 mg/L tricaine methane sulfonate (MS 222) before dissection. After the tissue was dissected, we froze the components quickly with liquid nitrogen and stored them in a −80 °C refrigerator for subsequent operations. The two populations used in the experiment were ZD (control group) and ZS (experimental group).

### 2.3. RNA Extraction

Total RNA was extracted from the tissue using TRIzol^®^ Reagent according to the manufacturer’s instructions (Invitrogen, Waltham, MA, USA). The genomic DNA was digested utilizing RNase-free DNase I (TaKara, Dalian, China) following the manufacturer’s protocols. Then, the RNA quality was measured using 2100 Bioanalyser (Agilent, Santa Clara, CA, USA) and quantified using ND-2000 (NanoDrop Technologies, Wilmington, DE, USA). High-quality RNA (OD260/230 ≥ 2.0, OD260/280 = 1.8~2.2, 28S:18S ≥ 1.0, RIN ≥ 6.5, >10 μg) was obtained for the sequencing library construction.

### 2.4. Library Preparation, and Illumina Hiseq Sequencing

RNA-seq transcriptome libraries were constructed using 1 μg of total RNA according to the TruSeq^TM^ RNA preparation kit from Illumina (San Diego, CA, USA). In short, we used oligo (dT) beads combined with polyA to separate the messenger RNA, and then a fragmentation buffer was used for fragmentation. The cDNA synthesis, end repair, A-base addition, and ligation of the Illumina-indexed adaptors were carried out according to Illumina’s scheme. Target fragments with cDNA of 200–300 bp were selected as library sizes on 2% Low Range Ultra agarose, and 15 PCR cycles were performed using Phusion DNA polymerase (NEB). Paired-end libraries were sequenced by Shanghai Biozeron Biothchology Co., Ltd. (Shanghai, China) with the Illumina HiSeq PE 2× 150 bp read length, after quantification with TBS380.

### 2.5. De Novo Assembly and Annotation

Trimmomatic use default parameters to trimming and quality controlled the raw data [31]. The clean reads (filtering out low-quality (Q ≤ 20) reads and adaptors) were obtained from the raw data according to the following criteria: shorter than 75 bp, N-base ratio over 10%, or mismatch/adapter sequences. Then, the clean data were used for RNA de novo assembly within Trinity [32]. Normally, amino acid sequences shape their spatial structure and determine their functional expression. Therefore, the more similar a protein is, the more similar its function will be. The assembled sequence then needed to be annotated functionally. Based on the principle of similarity, using BLASTX (translating a given nucleic acid sequence into a protein according to six reading frameworks and comparing it with sequences in the protein database), we referred to the Search Tool for the Retrieval of Interacting Genes (String) [33], NCBI protein nonredundant (NR), Kyoto Encyclopedia of Genes and Genomes (KEGG) and Clusters of Orthologous Groups of proteins (COG) [34] databases to find the protein with the highest sequence similarity to a given transcript(with cut-off E-values less than 1.0 × 10^−5^; the design of threshold values is of universal reference [35,36,37]) to retrieve their function and annotations. For unique assembled transcripts, we used the BLAST2GO (http://www.blast2go.com/b2ghome, accessed on 15 December 2021) [38] program to obtain Gene Ontology (GO) annotations and biological processes, molecular functions, and cellular components. KEGG (http://www.genome.jp/kegg/, accessed on 15 December 2021) [39] was utilized for metabolic pathway analysis.

### 2.6. Differential Expression Analysis and Functional Enrichment

To determine the differentially expressed genes (DEGs) between the different groups, the expression level of each transcript was calculated, referring to the reads per kilobase of exon per million mapped reads (RPKM) [40,41]. RSEM [42] was used to quantify gene and isoform abundance. EdgeR (empirical analysis of digital gene expression in R, http://www.bioconductor.org/packages/2.12/bioc/html/edgeR.html, accessed on 16 December 2021) [43] was used for differential expression analysis. The DEGs were identified according to the following criteria: (I). The logarithm of the fold change > 1; (II) the false discovery rate (FDR) < 0.05. To infer the function of DEGs, Goatools (https://github.com/tanghaibao/Goatools, accessed on 16 December 2021) and KOBAS (http://kobas.cbi.pku.edu.cn/home.do, accessed on 18 December 2021) were used, respectively, for GO functional enrichment and KEGG pathway analyses. It was considered that the DEGs were significantly enriched in GO terms when the multiple corrected *p*-value was <0.05 in Fisher’s test, according to the previous judgment methods [44]. At the same time, we used q-values to improve the statistical significance of the estimated error discovery rate (FDR). Because the number of GO Terms can be large, we analyzed only the first 30 terms (choosing the top ten for each process), and each comparison had a significant richness (from small to large in *p*-values).

## 3. Results

### 3.1. Data Quality Control and Unigene Analysis

The raw reads sequenced were in the range 38,312,462 to 51,005,574, and the clean reads obtained after quality control filtering ranged from 35,580,430 to 46,237,850 (Table 1). The lowest Q30 values of raw reads and clean reads were 91.07% and 94.62%, respectively (Table 1). The error was less than 0.025% (Table 1). After filtering out the low-quality and contaminated sequences, the mapping recorded that the length of mapped read ranges was from 31,443,076 to 37,913,141; the highest mapped rate was 93.13%, and the lowest was 82.00% (Table 1). Overall, the quality of the clean reads was suitable for subsequent analysis. De novo assembly produced 84,603 unique transcript fragments (unigenes), of which the average length was 872 bp and the N50 value was 1060 bp (Appendix A). The length distribution of the assembled unigenes gene is shown in Appendix A. Among 84,603 unigenes, at least one database had 30,677 annotated, including 30,057 (35.53%) of NR, 11,324 (13.38%) of GO, 23,563 (27.85%) of COG, 17,962 (21.23%) of KEGG, and 22,282 (26.34%) (Table 2).

### 3.2. Annotation and Function Analysis

All unigenes with *C. striata* identified were used for functional enrichment and classification analysis. Among them, 23,563 (27.85%) unigenes were classified into 25 different categories by COG annotation (Figure 1). After removing proteins with unclear characteristics (“only general functional clusters” and “unknown function” (columns ‘R’ and ‘S’ in Figure 1)), “replication, recombination and repair” (columns L) (451), “translation, ribosomal structure and biogenesis” (column J) (462) and “posttranslational modification, protein turnover, chaperones” (column O) (650) were listed as the top three functional clusters (Figure 1).

After the GO database was searched and compared, 11,324 unigenes were annotated to 63 GO terms, of which 27 GO terms clustered to biological process (BP), 21 GO terms clustered to cell component (CC), and 15 GO terms clustered to molecular function (MF) (Figure 2). Cellular processes (6508) and single-organism process (5223) were the largest subtypes of BP; cell (4271) and cell part (4208) had the highest enrichment in CC, and binding (5872) and catalytic activity (4385) were dominant in MF (Appendix A).

There were significant matches, for 17,962 annotated unigenes were distributed on 33 KEGG pathways (Appendix A). The global and overview map shows that the most genes (6745) are annotated into metabolism-related pathways, such as amino acid metabolism and lipid metabolism, and a total of 6270 genes are annotated in the Organismal Systems pathways, including the immune system, endocrine system, and digestive system. The rest–cellular processes, environmental information processing and genetic information processing–annotated 3395, 3608 and 2087 genes respectively.

### 3.3. Identification and Functional Analysis of Differentially Expressed Genes

Based on the value of transcripts RPKM, the gene expression peaks detected in these samples are basically similar (Figure 3a). According to the conditions |log^(fold change)^| ≥ 1 and false discovery rate (FDR) ≤ 0.05, a scatter plot was used to analyze the significantly changed genes. There were 9469 genes differentially expressed in ZD and ZS in Figure 3b. Among them, there were 5159 up-regulated and 4310 down-regulated genes. Inferring from the scatter plot, the Pearson correlation index was 0.9874. Nitrate stress treatment can lead to a biological stress response, which leads to the difference of expression levels between the control group and the nitrate treatment group. After nitrate treatment, a total of 8920 genes had significant changes; among them, 2949 genes were up-regulated, and 5971 genes were down-regulated (Appendix A).

From the perspectives of MF, BP, and CC, the DEGs in the comparison were analyzed for GO enrichment. Due to the large number of GO Terms, the first 30 terms were only analyzed choosing the top ten for each process, and each comparison had significant richness from small to large with the *p*-value. Among the first 30 GO terms analyzed, most DEGs were abundant in BP, followed by CC and MF. After the GO Terms on DEGs, three terms were the most significant: positive regulation of lipid storage (GO:0010884), lipid localization (GO:1905954), and lipid storage (GO:0010883) (Appendix A and Figure 4). They were significantly enriched in the total gene proportions with 0.13%, 0.15%, and 0.14% components, respectively (Appendix A). The most significant ones in biological process, the cellular component, and molecular function were positive regulation of lipid storage (GO:0010884), blood microparticle (GO:0072562) and inhibitors of enzyme activity (GO:0004857). The percentages of total genes for blood microparticle (GO:0072562) and inhibitors of enzyme activity (GO:0004857) were 0.39% and 0.59%, respectively (Appendix A).

To clarify the possible functional status of the KEGG pathway, in the assembled transcripts, the first 30 pathways for each comparison were analyzed. The DEGS after nitrate treatment matched 315 different KEGG pathways (Appendix A). The top terms with the most significant enrichment of DEGs in KEGG were complement and coagulation cascades and Staphylococcus aureus infection (Figure 5).

## 4. Discussion

Stress is defined as a series of physiological events that occur when an organism tries to re-establish the norms of homeostasis in the face of a perceived threat [45]. It refers to the physiological and efficient response of organisms to sudden and drastic changes in the environment. Generally, it is related to the characteristics of the species itself, but in aquaculture, it can be continuously optimized through breeding to improve the ability of the species to face an adverse environment. Whether in aquatic animals or plants, different strategies are used to adapt to change in the nitrate environment for nitrate stress. In particular, the increase of nitrite concentration may cause the acute toxicity of aquatic animals, including energy metabolism damage and tissue damage. In this study, we analyzed the full transcriptome expression and differential gene expression of *C. striata* in normal culture conditions and transfer to nitrate-stressed environments. After sodium nitrate treatment, transcriptome analysis demonstrated that the expression levels of 8920 genes were significantly changed, of which 2949 up-regulated genes and 5971 down-regulated genes were found (Appendix A). The GO and KEGG analysis of differential genes showed significant changes, which indicated that nitrate stress had a significant effect on the liver of *C. striata*. In this study, we also found that *C. striata* under nitrate stress can respond to this environmental change in various ways to avoid being harmed in this environment. Our results were consistent with the research of others, demonstrating that under high nitrate concentration, fish tissues and organs will undergo various regulatory changes to adjust themselves to sudden environmental changes and ensure the survival of the body [46]. Moreover, this regulatory response is complex and multifaceted.

After nitrate treatment, the total gene expression changed significantly (Appendix A). In this study, C. striata were treated with nitrates for the first time, transcriptome sequencing was performed on the liver, and the differentially expressed genes were analyzed. Through GO enrichment analysis of DEGs, the most significant enrichment levels were found to be the positive regulation of lipid storage (GO:0010884), lipid localization (GO:1905954), and lipid storage (GO:0010883) (Figure 4). In the 30 GO enrichment analyses, the focus was on the biological process, indicating that the expression of different genes in the liver was mainly related to lipid changes. Lipid is one of the three major nutrients in the body essential to life activities. It can participate in the synthesis and decomposition of substances and energy through the tricarboxylic acid cycle to provide energy for the body. Metabolites can also participate in the synthesis of other substances in the body (such as hormones, ligands that activate enzymes, signaling molecules), maintain the normal physiological function of the body, or transmit signals to adjust the cascade reaction to accelerate the regulatory process and metabolic regulation activities. Lipids can provide essential fatty acids for the fish body. When the fish body lacks essential fatty acids, it will significantly affect the aquaculture production and even reduce the immunity and anti-stress ability of the fish [47]. Therefore, in this study, *C. striata* may respond by increasing lipids under abiotic stress. The GO enrichment analysis of the cellular component mainly focuses on blood microparticles (GO:0072562). Microparticles (MP) in the blood are considered to be small particles that are shed in the blood circulation by cell activation or apoptosis due to stimulation [48]. Although polysulfonic acid mucopolysaccharide has long been considered cell debris, it was recently thought that it is a reflection product of cell stimulation, activation, and degeneration/apoptosis [49,50,51]. In previous studies, it was found that blood microparticles may participate in the activation of the protein C resistance factor to induce blood coagulation and other functions [52]. The results of this study suggest that *C. striata* may be resistant to nitrate stress through blood microparticles. The GO enrichment analysis of molecular function mainly focuses on inhibitors of enzyme activity (GO:0004857). During the growth of organisms, enzymes in the organism regulate these various reactions to face changes and growth in the environment. In the liver of fish, various enzymes will play a role in the face of pathological damage to resist such damage. Any stress factor may cause fish to suffer from internal physiological imbalance through disordered hormone and enzyme functions and changes in certain blood characteristics, thereby affecting growth [53]. Our research shows that when *C. striata* are under abiotic influences (nitrate stress), they can respond to this nitrate stress by regulating enzyme activity.

The KEGG database is a collection of all fully sequenced genomes and gene catalogs of partial genomes, with the latest gene function annotations [54]. The KEGG pathway database contains previously uploaded and published reliable information on identified metabolic pathways and molecular interactions in regulatory networks [55]. The DEGs in the liver of *C. striata* treated with nitrate in this study matched 315 different KEGG pathways. We found that the most significant pathways in KEGG are photosynthesis, complement and coagulation cascades, Staphylococcus aureus infection, asthma, and antigen processing and presentation (Figure 5). There are a total of 264 genes in the coagulation cascades and complement pathway, of which 71 DEGs were annotated into this pathway (Appendix A). Previously, it was thought that the complement and coagulation cascade were two separate systems, but recent studies have shown that the two reactions can often be activated simultaneously and cooperate with each other [56,57]. The complement system is defined as the composition of more over 35 soluble plasma proteins, which play an important role in adaptive immunity and innate immunity [58]. In the fish immune system, blood clotting traps “foreign” threats that have invaded the blood vessels in the agglutinated clots and promotes the innate immune system by increasing vascular permeability and the instinctive phagocytosis capacity of phagocytes [59]. In the study of Yin et al., it was found that the complement and coagulation cascade pathways may play an important role in the early immune defense of bacterial infection [60]. Zhang et al. [59] also showed similar results in the study of *Vibrio anguillarum* infection of *Cynoglossus semilaevis*. Since nitrate in the water environment reaches a certain concentration, it will cause fish poisoning and death, so we speculate that the way of complement and coagulation cascades may be related to the deterioration of the water environment. There was a total of 180 genes in the Staphylococcus aureus infection pathway, of which 54 DEGs were annotated into this pathway (Appendix A). Based on such enrichment results, we speculated that due to the toxic effect of nitrate, the immune function of fish was greatly reduced, and the defense ability against bacterial infection was weakened. At the same time, it is easier for bacteria to enter the fish, so the body activates the relevant immune system to carry out immune work to protect itself. At the transcriptome level, we were able to enrich differentially expressed genes related to the Staphylococcus aureus infection pathway. In this study, the DEGs of *C. striata* treated with nitrate were enriched differently (GO enrichment and KEGG enrichment). Compared with KEGG analysis, GO analysis produced fewer enrichment pathways. We believe that our findings will help analyze gene expression of other fish under nitrate stress.

## 5. Conclusions

In this study, the reassembly of next-generation sequencing data was used to study the effects of toxicity stress (nitrate treatment) on the liver of *C. striata*. Nitrate treatment caused significant changes in a total of 8920 genes, of which 2949 genes were up-regulated and 5971 genes were down-regulated. These differential genes were enriched and analyzed, among which 236 different pathways were matched by GO enrichment analysis, and 315 different pathways were matched by KEGG enrichment analysis. Transcriptome analysis identified the DEG in the response of nitrates in the liver of black sea bass, and bioinformatics analysis linked many DEGs to regulatory pathways such as lipid metabolism, immune response, and inflammation. Our data can provide differential expression data for the stress regulation response of black sea bass under nitrate stress, lay a foundation for exploring biological cell pathways under abiotic stress, and provide data support for screening candidate genes for stress resistance. The exploration of toxicity-tolerant genes can confirm that certain genes can help black sea bass survive under toxicity stress conditions, which can increase production in aquaculture in the face of abiotic stress.

## Figures and Tables

**Figure 1 genes-14-01440-f001:**
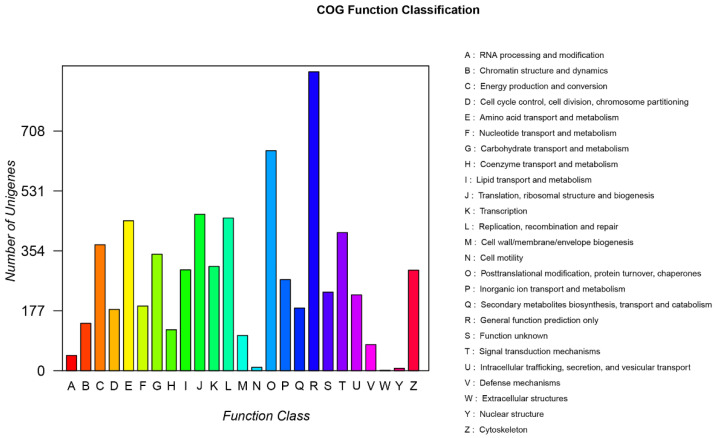
COG annotation of all identified unigenes from *C. striata* with and without nitrate treatment.

**Figure 2 genes-14-01440-f002:**
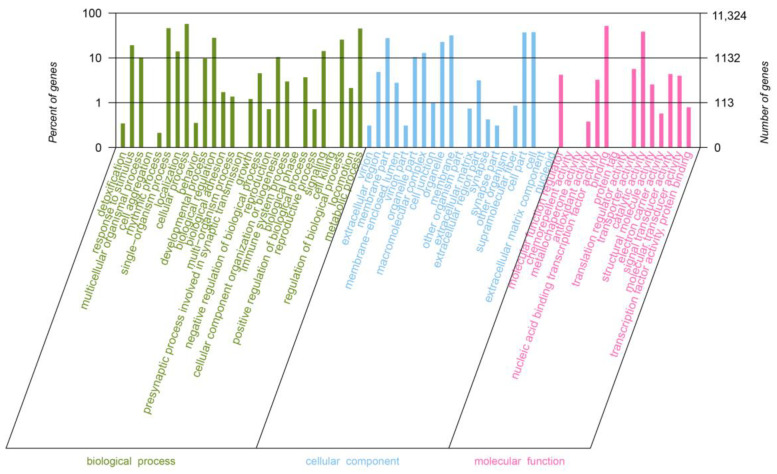
GO annotation of all identified unigenes from *C. striata* with and without nitrate treatment.

**Figure 3 genes-14-01440-f003:**
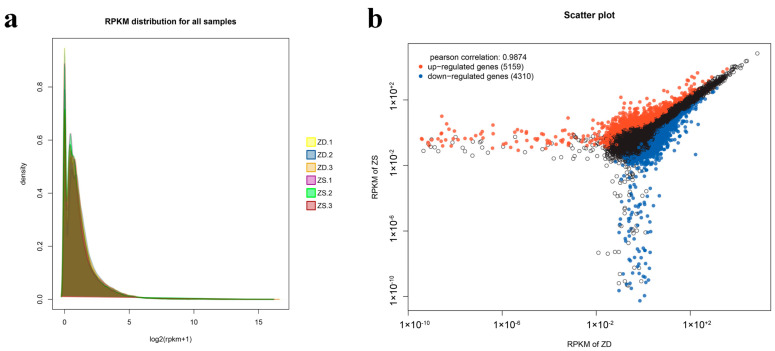
(**a**) Expression level (reads per kilobase of transcript per million reads mapped scores), distribution map, and (**b**) visualized scatter plot of differentially expressed genes (black dots indicate genes that are not significantly different).

**Figure 4 genes-14-01440-f004:**
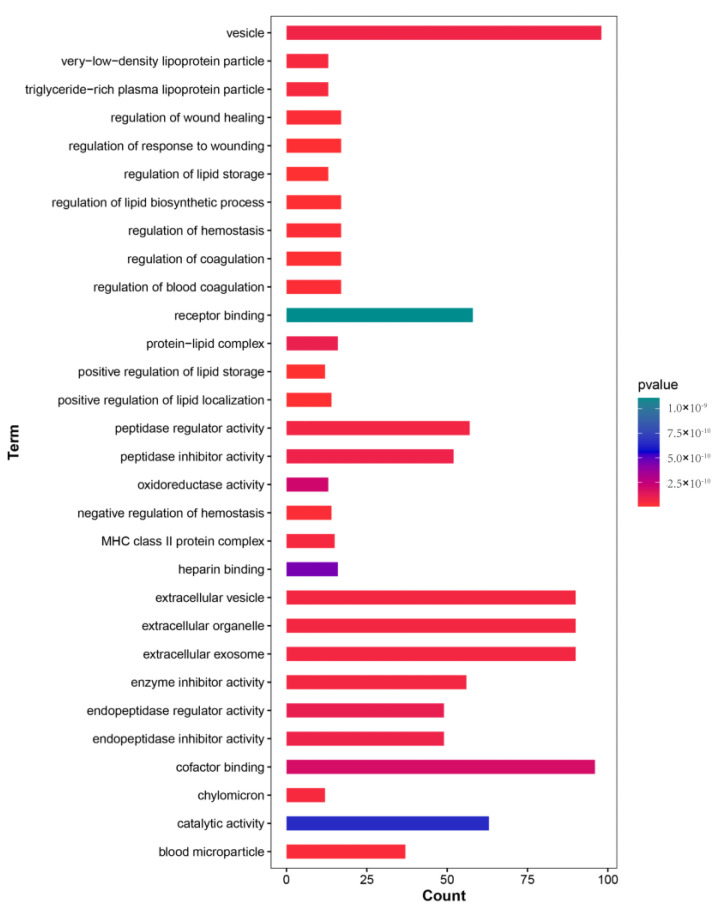
Top gene ontology-enriched terms.

**Figure 5 genes-14-01440-f005:**
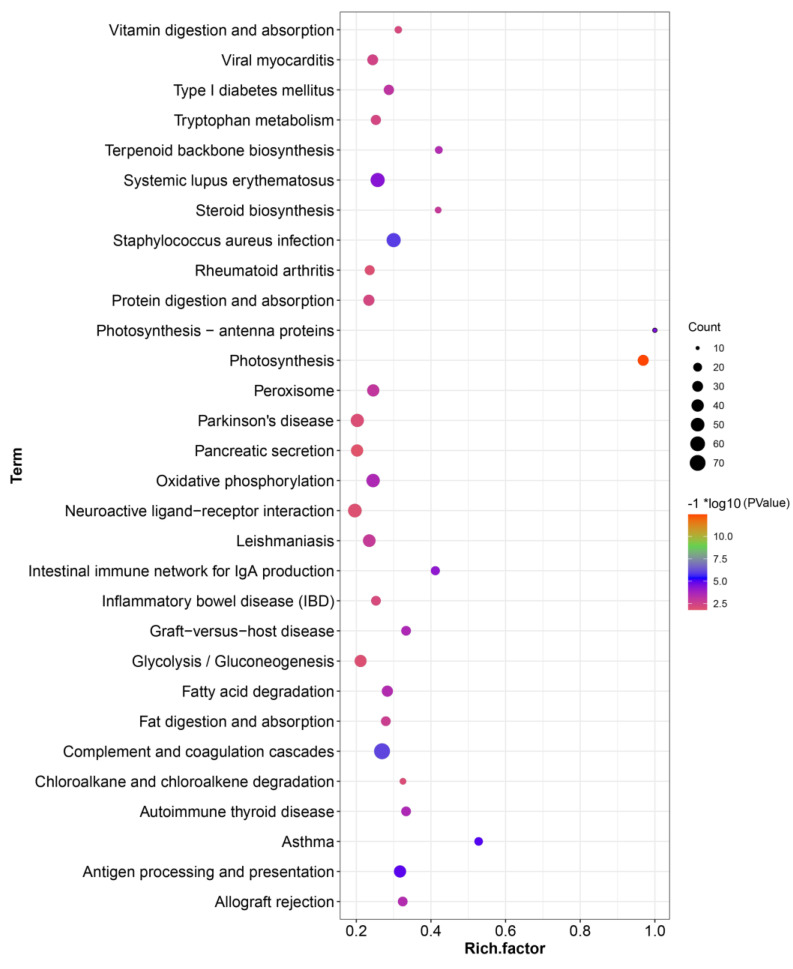
Top pathways enriched in the Kyoto Encyclopedia of Genes and Genomes analysis. A large and rich factor indicates a high degree of enrichment.

**Table 1 genes-14-01440-t001:** Summary of the transcriptome assembly of *C. striata*. Control (ZD) and nitrate-treated (ZS) raw read filters.

Sample	Raw Reads	>Q30	Clean Reads	>Q30	Error%	Mapped Reads	Mapped Rate (%)	GC%
ZD-1	44,436,926	92.82	41,499,506	94.62	0.0244	34,537,922	83.22	50.61
ZD-2	40,784,182	93.45	37,947,644	95.02	0.024	31,443,076	82.86	50.69
ZD-3	38,312,462	93.09	35,580,430	94.78	0.0242	29,808,501	83.78	50.75
ZS-1	45,948,888	92.15	41,841,058	95.32	0.012	34,773,392	83.11	50.07
ZS-2	46,607,084	92.15	42,642,714	95.3	0.012	35,294,478	82.77	50.27
ZS-3	51,005,574	91.07	46,237,850	94.85	0.0124	37,913,141	82.00	49.25

**Table 2 genes-14-01440-t002:** Annotation of unigenes against five databases.

Database Name	Total Unigenes	Annoted Unigenes	Percent
NR	84,603	30,057	35.53%
GO	84,603	11,324	13.38%
COG	84,603	23,563	27.85%
KEGG	84,603	17,962	21.23%
SWSS	84,603	22,282	26.34%
In all databases	84,603	7,293	8.62%
At least one database	84,603	30,677	36.26%

## Data Availability

The sequences have been submitted to NCBI GenBank under the BioProjrct PRJNA685629. The raw sequencing reads of RNA are available at SRA (SRR13271206; SRR13271205; SRR13271204; SRR13271203; SRR13271208; SRR13271207).

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
