# Peer review of "Transcriptomic Analysis of Liver Tissue of Black Sea Bass (Centropristis striata) Exposed to High Nitrogen Environment"

_genes, 2023, doi:10.3390/genes14071440_

Round 1

Reviewer 1 Report

Manuscript ID: genes-2474224

The manuscript “Transcriptomic analysis of of liver tissue of black sea bass (Centropristis striata) exposed to high ammonia nitrogen environment” aims to explore by RNA-seq technology the set of genes differentially expressed in a nitrate highly-concentrated breeding environment of C. striata

I believe the study is relevant and presents some novelty for the aquaculture field. However, the experimental design and sampled groups are weak, the type of analysis too general, and the conclusions not suitable.

Some effort has to be done in order to improve the text clarity and quality of the manuscript .

Some major point raised were:

How is the status concerning Centropristis striata sequenced genome? How much of their genome is known? Is it possible to do genome annotation for the sequenced genes with high reliability?

Please elaborate on why genes associated with salt tolerance under nitrate stress were intended to be characterised. Are there specific markers for nitrate stress? Please explain why an RNAseq study works better for this experimental design (since it seems a kind of target analysis). Also, why in this environment would the liver be the most affected organ?

Some other noticed details include:

Title and abstract: Authors use the term “ammonia nitrogen”; however, since ammonia is one of several forms of nitrogen that exist in aquatic environments, wouldn't it be better to simply use nitrogen/nitrogen stress when referring to the main stressor? (Also correct the title, ‘of’ appears twice).

Abstract: “GO and KEGG analysis showed that the most significant enrichment genes were associated with the genes that are related to pathways such as…” This part is very confusing. First, the GO terms are the ones that get ‘enriched’ by DEGs, and the GO terms are those related to pathways. Please, rewrite. Still, in the abstract, if the genes obtained in GO and KEGG enrichment analysis were mostly related to the immune and inflammatory response of fish why GO terms itself related to immune response were not enriched? What kind of bioinformatics analysis pointed to those conclusions?

Keywords: To increase the spread of the research, please use keywords different from the title.

Introduction:

Line 28: Instead of describing with personal arguments such as “delicious meat”, state that it is a highly consumed species, for example.

40-42: “With the growth of aquaculture in the direction of ecological sustainability, the study with reducing the adverse effects of aquaculture activities on the environment has gradually become a research hotspot for domestic and foreign scholars”. I don’t know how important that justification is since the main interest here is to provide a less harmful environment for the creation of this species for aquaculture, not necessarily looking forward to environmental preservation.

“Therefore, the purpose of this research is to find resistance genes through RNA-seq technology and solve a series of problems caused by the increase of nitrate concentration in C. striata breeding. At the same time, our data additionally provides valuable genomic resources for C. striata.” How does the RNAseq study help to find resistance genes? In case one can identify those genes, how could such knowledge be applied?

Methodology:

The study consists of one replicate? 6 animals, 3 controls and 3 exposed? It should be performed at least in triplicate.

How did the authors get to the concentration of 450 mg/L nitrate? Is this a real concentration in high nitrate environment of aquaculture?

Results:

Line 205: It may not be very clear what is the salt treatment mentioned in “significant changes after salt treatment”. Throughout the manuscript I am not sure if “salt stress” is the best way to describe the experimental condition (just in the conclusion section authors wrote “salt stress (nitrate treatment)”, that could appear way earlier to help understanding.)

Lines 211-217: Please rewrite to increase clarity. If any special treatment has been done in the data, explain in the methodology.

Please, pay attention to scientific names always in italic.

Discussion:

Lines 255-256: “to perform transcriptome analysis of the differential expression of genes in the liver”...

GO and KEGG are quite broad analyses pointing to very unspecific pathways. How did the author use those results to specifically answer the scientific question (find resistance genes through RNA-seq technology and solve a series of problems caused by the increase of nitrate concentration in C. striata breeding)? I do not see clearly how the study helps on that, please elaborate. I would say the results poorly helped to reach the research aim.

Conclusion:

The aim presented in the conclusion section “...reassembly of next-generation sequencing data was used to study the effects of salt stress (nitrate treatment) on the liver of C. striata” is considerably different from the one presented in the end of the intro. Personally the last one is better; also because the authors did not present any “resistance genes found” .

Line 323: “bioinformatics analysis linked many DEGs to immune response and inflammation”...I do not think the text strongly supports such a conclusion, please I suggest editing this matter.

What are the potential candidate genes mentioned in the conclusion? (“Our data on DEGs can also provide potential candidate genes for the functional analysis of black sea bass survival under salt stress.”)

Overall, I think the study does not corroborate the conclusions presented.

Reviewer 2 Report

Manuscript by Bingjian Liu et al. ‘’Transcriptomic analysis of liver tissue of black sea bass (Centropristis striata) exposed to high ammonia nitrogen environment’’ investigates molecular responses to elevated ambient ammonia nitrogen in a commercial fish species. The data provide valuable genomic information for C. striata. Most of the DEGs were found to be related to the immunity and inflammatory response.

Line 2, the title. Typo error – duplicated ’of’

Line 16 and throughout, You mention “nitrate stress” as a synonym for salt stress. However, salt stress (and salinity tolerance and other related phenomena) is more concerned with osmotic imbalance/shock. Please find appropriate words to describe "nitrate stress" that emphasize the toxicity rather than the osmotic effect of nitrogen compounds.

Line 19, I suggest replacing "enzyme inhibitor activity" with "enzyme inhibitors" or "inhibitors of enzyme activity" (as the activity is inherent in the enzyme, not the inhibitor)

Line 67, The phrase ‘The transcription information of organisms’ seems to be better replaced by ‘The transcriptome of an organism’

Lines 85-102, point 2.2, the material is poorly characterized, key characteristics are missing, including size, age of fish, whether they were kept individually or in groups, water temperature, oxygenation, volume of water replacement, etc.

Line 91, "the experiment was divided into experimental group" may be better to use "sample" or "fish" instead of "experiment".

Line 95, The phrase ‘During the experiment, keep aerating, feed…’ should be checked for the subject and the correct verb form (e.g., During the experiment C. striata was fed normally, aquaria water maintained aerated, etc.)

Line 101, the abbreviations used for the fish groups "ZD (control group) and ZS (experimental group)" seemed incomprehensible (meaning neither species nor treatment), couldn't you explain them or replace them with more predictable ones?

Line 173, please unify the description of the clusters given in the text and in fig. 1 ("only general functional clusters" and "general function prediction only")

Line 238, inappropriate expression "a person" regarding fish

Lines 277-278, the authors can only speculate on the role of blood microparticle accumulation in nitrate-treated fish; there is no evidence whether "C. striata may resist salt stress through blood microparticles" or whether the multiple microparticles originate from damaged blood cells.

Round 2

Reviewer 1 Report

Thanks for providing an updated version of the manuscript where considerable improvements were made. I really appreciate it.

I just would add a few comments; concerning the status of the Centropristis striata sequenced genome it is important that authors make it clear in the text how was the annotation made and how precise it is, even if based on other experimental models or comparisons, in order to guide the readers in a more accurate methodological description. For the enrichments analysis with (GO and KEGG) [“Through GO and KEGG analysis of differentially expressed genes, it was found that significantly enriched genes were associated with”], I think that differentially expressed genes from sequencing are used to enriched GO terms and KEGG pathways; so it would the best to describe those parts as “it was found that significantly enriched GO terms [or] ontologies were associated with…”. Overall, since the study provided meaningful inferences that may support further studies but still have a few limitations, it is very important to write the conclusions in a down-to-earth view, i.e., avoid broad extrapolations and concentrate only on the findings in a straightforward manner.
